# Prediction of excess pregnancy weight gain using psychological, physical, and social predictors: A validated model in a prospective cohort study

Sarah D. McDonald[1,2,3☯]*, Zhijie Michael Yu[1☯], Sherry van Blyderveen[4☯], Louis Schmidt[5☯], Wendy Sword[6☯], Meredith Vanstone[7☯], Anne Biringer[8☯], Helen McDonald[9☯], Joseph Beyene[3☯]

1 Division of Maternal-Fetal Medicine, Department of Obstetrics and Gynecology, McMaster University, Hamilton, Ontario, Canada, 2 Department of Radiology, McMaster University, Hamilton, Ontario, Canada, 3 Department of Health Research Methods, Evidence & Impact, McMaster University, Hamilton, Ontario, Canada, 4 Eating Disorders Program at Homewood Health Centre, Guelph, Ontario, Canada, 5 Department of Psychology, Neuroscience & Behaviour, McMaster University, Hamilton, Ontario, Canada, 6 School of Nursing, McMaster University, Hamilton, Ontario, Canada, 7 Department of Family Medicine, McMaster University, Hamilton, Ontario, Canada, 8 Ray D. Wolfe Department of Family Medicine, Mount Sinai Hospital, Toronto, Ontario, Canada, 9 Midwifery Education Program, McMaster University, Hamilton, Ontario, Canada

☯ These authors contributed equally to this work.
* mcdonals@mcmaster.ca

**Data Availability Statement:** All relevant data are within the manuscript and its Supporting Information files.

## Abstract

### Objective

To develop and validate a prediction model for excess pregnancy weight gain using early pregnancy factors.

### Design

Prospective cohort study

### Setting

We recruited from 12 obstetrical, family medicine, and midwifery centers in Ontario, Canada

### Participants

We recruited English-speaking women with singleton pregnancies between $8^{+0}$–$20^{+6}$ weeks. Of 1296 women approached, 1050 were recruited (81%). Of those, 970 women had complete data (970/1050, 92%) and were recruited at a mean of 14.8 weeks.

### Primary outcome measure

We collected data on psychological, physical, and social factors and used stepwise logistic regression analysis to develop a multivariable model predicting our primary outcome of excess pregnancy weight gain, with random selection of 2/3 of women for training data and 1/3 for testing data.

**Funding:** This research was supported by a Canadian Institutes of Health Research grant # MOP-142253 (http://www.cihr-irsc.gc.ca/e/193. html). SDM is supported by a Tier II Canada Research Chair Sponsor Award #950-229920 (https://www.chairs-chaires.gc.ca/home-accueil-eng.aspx). AB is supported by the Ada Slaight and Slaight Family Maternity Care Directorship. The funders had no role in study design, data collection and analysis, decision to publish, or preparation of the manuscript.

**Competing interests:** The authors have declared that no competing interests exist.

## Results

Nine variables were included in the final model to predict excess pregnancy weight gain. These included nulliparity, being overweight, planning excessive gain, eating in front of a screen, low self-efficacy regarding pregnancy weight gain, thinking family or friends believe pregnant women should eat twice as much as before pregnancy, being agreeable, and having emotion control difficulties. Training and testing data yielded areas under the receiver operating characteristic curve of 0.76 (95% confidence interval, 0.72 to 0.80) and 0.62 (95% confidence interval 0.56 to 0.68), respectively.

## Conclusions

In this first validated prediction model in early pregnancy, we found that nine psychological, physical, and social factors moderately predicted excess pregnancy weight gain in the final model. This research highlights the importance of several predictors, including relatively easily modifiable ones such as appropriate weight gain plans and mindfulness during eating, and lays an important methodological foundation for other future prediction models.

## Introduction

Half or more of women in the United States, [1] Europe, [1] and Canada, [2] and 37% of women in Asia [1] exceed the guidelines from the Institute of Medicine for weight gain during pregnancy [3] which were also adopted by Canada, [4] and a number of other countries. [5, 6] Gaining in excess of guidelines significantly increases infant risk of high birth weight [2] and maternal risks of hypertension, [7] diabetes, [8] caesarean section, [1] and postpartum weight retention. [9]

Hundreds of studies have examined factors associated with weight gain in pregnancy, [10–12] but to date, validated models are lacking. Interventions to prevent excess pregnancy weight gain have been largely unsuccessful, [13–15] or with minimal improvement. [16] Multiple recent meta-analyses of interventions have indicated one potentially fruitful area for study is examination of psychological factors influencing weight gain. [17–19] In response, we undertook a systematic review of psychological factors associated with excess pregnancy weight gain, [20] identifying a number of novel areas for exploration in the four broad psychological domains: 1) cognition (e.g., normative factors), 2) affect (e.g., pregnancy-related anxiety [21]), 3) personality (e.g., impulse control, [22] perfectionism, [23] emotion suppression, [24] and the Big 5 Personality Factors [25], a standard classification) and 4) behavior (e.g., emotional eating, [26] night eating). In addition to psychological factors, our pilot study found that planning to gain weight above the recommendations was associated with an increased likelihood of developing excess pregnancy weight gain compared with planning to gain weight within the recommendations. [27] Other studies also reported that some physical factors, including parity [10] and prepregnancy BMI, [28] might play a role in excess pregnancy weight gain. Identifying women at high risk of excess pregnancy weight gain relatively early in pregnancy may allow interventions targeting those at high risk and reduce unnecessary interventions in women at low risk.

The aim of this study was to develop and validate a predictive model of excess pregnancy weight gain using psychological, physical, and social determinants of excess pregnancy weight gain collected in early pregnancy. We hypothesized that a model with the combination of

psychological, sociodemographic, behavioural, and physical factors would predict excess pregnancy weight gain.

## Materials and methods

We followed the "Transparent Reporting of a Multivariable Prediction Model for Individual Prognosis or Diagnosis: the TRIPOD statement", [29] an evidence-based set of recommendations for reporting prediction studies. It standardizes the reporting of prediction modeling studies, thus aiding their critical appraisal, interpretation and uptake by potential users. [29] We reported our findings according to the *Strengthening the Reporting of Observational Studies in Epidemiology (STROBE)* guideline. [30] The research project was approved prior to study initiation by the 1) Hamilton Research Ethics Board (REB #13–021), 2) Mount Sinai Hospital Research Ethics Board, 3) University Health Network Research Ethics Board, 4) Ottawa Health Sciences Network Research Ethics Board, 5) Lakehead University Research Ethics Board, and 6) Thunder Bay Regional Health Sciences Centre Research Ethics Board.

We recruited women in Ontario, the province with the largest proportion of births in Canada (139 999 of a total 376 291 in 2017). [31] We included large and smaller urban centers from the five regions of Ontario. [32] We included 12 clinics from the three main groups of pregnancy health care providers: obstetricians, family physicians, and midwives. [33] We used rolling recruitment of centres and patients were recruited from October 2015 to April 2017.

We included pregnant women with a live, singleton fetus from 8 weeks + 0 days to 20 weeks + 6 days gestation who could read and write English. Eligible participants planned to give birth at the same centre as they were recruited, to facilitate data collection on outcome.

We excluded the following conditions: i) twins or higher order multiples as weight gain recommendations differ, [3] ii) a fetus with a known lethal anomaly, a fetal demise, or a termination of pregnancy after enrollment, and iii) maternal pathological conditions that severely impact weight gain due to extreme diet (e.g., bariatric surgery, [34] anorexia, [35, 36] and bulimia [35]).

We aimed to recruit consecutive women in early pregnancy. For feasibility, we collected recruitment rates over the course of the first two weeks of recruitment. Participants provided written, informed consent before taking part. We used rolling recruitment at various centres between 2015–2017.

Our a priori sample size calculation was based on the standard rule of "10 events per variable" in the regression model, [37] although the more lenient five events per variable in the regression model is sometimes considered to allow additional factors to be included in the regression model, [38] resulting in the inclusion of nine predictor variables. Informed by data from our pilot study, [39] in which 50% of women gained above the guidelines and 50% within or below, we required approximately 340 women with excess pregnancy weight gain. We increased the sample size for missing data, based on our pilot study, by 3% for incomplete questionnaires and 10% loss to follow-up. Finally, we increased the sample size to account for model validation using a 2/3 to 1/3 ratio for training data to testing data. Hence, we estimated we would require approximately a total of 1042 participants at initial recruitment, incorporating the 10% loss to follow.

The questionnaire was developed based on: 1) our systematic review of the literature on psychological factors associated with weight gain in pregnancy [20] and 2) a pilot study [39] using existing, validated scales wherever possible. Five individuals with expertise (obstetrician, weight psychologist, research psychologist, perinatal nurse, and midwife) assessed the questionnaire for content validity.

We assessed previously unexplored factors in pregnancy [20] (S1 File). We also explored factors even if they had previously been evaluated [20] if they were believed to be important, including: 1) cognition (target weight gain, [40] weight attitudes, [41] body image, [42] self-efficacy, [41] weight locus of control, [41] Barriers to Healthy Eating), [43] 2) affect (depression, anxiety [21]), 3) personality (impulse control, [22] perfectionism, [23] emotion suppression, [24] and the Big 5 Personality Factors [25]), and 4) behaviour (dietary restraint, [44] diet, [45] physical activity, [46] sleep, smoking, eating in front of a screen) (questionnaire included in S2 File).

Additionally, the questionnaire collected sociodemographic determinants (e.g. maternal age, education, income) and physical determinants of excess pregnancy weight gain, including body mass index (BMI) and number of previous pregnancies. We also collected data on the participants' recollection of the health care providers' recommendations on GWG, including their recommended first trimester weight gain, total GWG, and planned GWG.

Research staff abstracted outcomes from the Antenatal Record forms mandated by the Ministry of Health, using a piloted data collection form. Height is recorded at the start of the pregnancy and weight at each antenatal visit (97% of visits) [47] on the Antenatal Records. We calculated total pregnancy weight gain by subtracting pre-pregnancy weight from the final measured weight, with both obtained from the Antenatal Records.

Our primary outcome was total pregnancy weight gain. We used the 2009 Institute of Medicine (IOM) guidelines [3] also adopted by Health Canada [4] on GWG to categorize women's weight gain as below, within or above recommended. Because GWG is associated with duration of pregnancy, we accounted for gestation age in our calculations. If a measured weight was not available on the antenatal records for the first trimester weight, as per the Institute of Medicine guidelines, [3] we assumed a 2 kg weight gain in the first trimester, and subtracted this amount from the total reported weight gain to obtain weight gain during the second and third trimesters of pregnancy. [3, 48] We then subtracted 13 weeks for the duration of first trimester from the gestational age at birth to obtain the number of weeks in the remainder of the pregnancy. We compared this weight gain in the remainder of the pregnancy (i.e. 2nd and 3rd trimesters) to the IOM's recommendations for pregnancy weight gain during this period, accounting for women's pre-pregnancy BMI using the World Health Organization cutoffs for underweight (BMI $< 18.5$ kg/m$^2$), normal weight ($18.5 \leq$ BMI $< 25$ kg/m$^2$), overweight ($25 \leq$ BMI $< 30$ kg/m$^2$) or obese (BMI $\geq 30$ kg/m$^2$) [49] as per the guidelines. [3]

We compared characteristics in women whose total pregnancy weight gain was above *versus* within or below the guidelines using univariate logistic regression. We used Spearman correlations to examine collinearity between variables. For those pairs with bilateral Spearman correlation coefficients $\geq 0.70$ or $\leq$ -0.70, we retained the most psychologically and biologically relevant variable. We addressed missing data with multiple imputations using the fully conditional specification method to create ten imputed data sets [50] with PROC MI in SAS. [51, 52] We calculated the means of the ten imputed values and rounded the means to the nearest integers of categorical variables and decimal values of continuous variables. We randomly split the data into training (67%) and testing (33%) data sets for the prediction model development and its validation.

We employed stepwise logistic regression for the selection of important predictors for computing the predicted probability of excess gestational weight gain, using a p value of <0.10 for entry into the model, as defined by the likelihood ratio test statistic. [53] We retained variables in the prediction model if p value < 0.05. We assessed model fit using the Hosmer and Lemeshow goodness-of-fit test. [53] To evaluate the performance of the prediction model for differentiating excess gestational weight gain among our participants in the training sample, we first calculated the area under the receiver operating characteristic curve. [54] Next, we calculated

the calibration slope. Discrimination and calibration were performed with the validation sample as well to validate the performance of the prediction model by using the parameters of the selected predictors derived from the training sample. We examined differences in area under the curve (AUC) between the training and validation data sets according to the method of DeLong et al. [55] We performed sensitivity analyses by restricting the study sample to women who were aged 20 years and older and separately to women who gained weight within and above the recommendations. We used SAS 9.4 software (SAS Institute, Cary, North Carolina) to perform data management and statistical analysis.

## Results

Among 1296 approached women, 1050 (81%) consented to participate. The main reasons for exclusion were miscarriage after recruitment, gestational age above the recruitment window (ending at 20 weeks and 6 days), and outcome unavailability. We found that missing data were not related to either major characteristics of our study participants, including maternal age, gestational age of baseline survey, and prepregnancy BMI, or the study outcomes (S3 File). We had complete outcome data on 970 women (92%, Fig 1).

The mean maternal age was 30.5 years and the mean gestational age at recruitment and completion of the baseline questionnaire was 14.8 weeks (Table 1). Just over half of participants were nulliparous. Majority of participants self-identified as white, with relatively high rates of being married or common law, high levels of educational attainment, and high household income. Approximately half of the women had a normal pre-pregnancy BMI, with the remaining evenly split between overweight and obese. More than half (55%) of women had a total weight gain exceeding the guidelines, 29% gained weight within the recommendations, and 16% gained below. Missing data ranged from 0.1 to 9.6% and were less than 3.2% for most variables (S4 File). The mean first trimester weight gain measured between 12–13 weeks of gestation was 1.83 ± 3.3 kg.

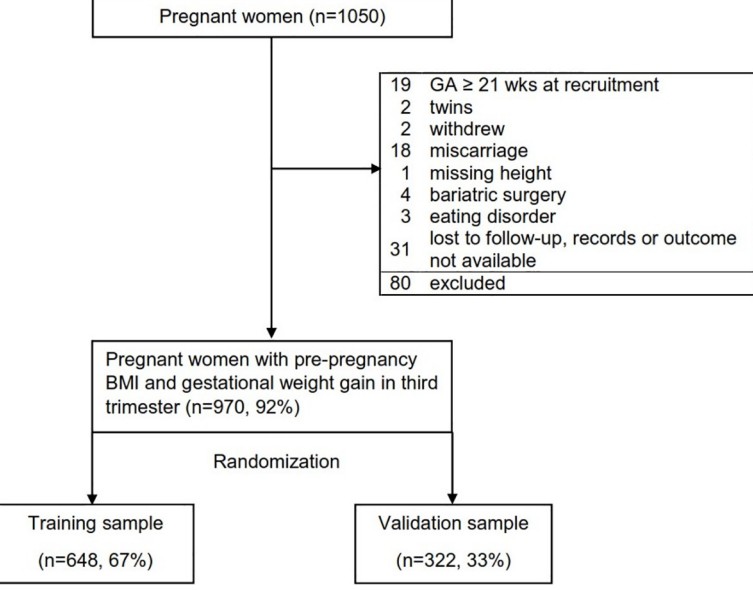

**Fig 1. Participant flowchart in prospective cohort to develop a prediction model of excess pregnancy weight gain.** BMI, body mass index; GA, gestational age; n, number; wks, weeks.

**Table 1. Baseline characteristics in prospective cohort to develop a prediction model of excess pregnancy weight gain.**

| Characteristic | (total n = 970) | |
|---|---|---|
| **Maternal age in years, Mean (SD)** | 30.5 | (4.9) |
| **Maternal age in years, Median (IQR)** | 31.0 | (27.0 to 34.0) |
| **Gestational age at recruitment, weeks, Mean (SD)** | 14.8 | (3.4) |
| **Gestational age at recruitment, weeks, Median (IQR)** | 15.0 | (12.3 to 17.6) |
| **Race, n (%)** | | |
| White | 736 | (75.9) |
| Non-white | 231 | (23.8) |
| Not reported/Unknown | 3 | (0.3) |
| **Marital status, n (%)** | | |
| Married, common-law, or living with a partner | 898 | (92.6) |
| Single, divorced, or widowed | 70 | (7.2) |
| Not reported/Unknown | 2 | (0.2) |
| **Education, n (%)** | | |
| Some high school or less | 70 | (7.2) |
| Completed high school | 56 | (5.8) |
| Community college or technical school (some or completed) | 286 | (29.5) |
| Undergraduate university (some or completed) | 319 | (32.9) |
| Graduate degree | 238 | (24.5) |
| Not reported/unknown | 1 | (0.1) |
| **Household income, n (%)** | | |
| < $10 000 | 19 | (2.0) |
| $10 000 - $19 999 | 42 | (4.3) |
| $20 000 - $39 999 | 91 | (9.4) |
| $40 000 - $59 999 | 114 | (11.8) |
| $60 000 - $79 999 | 136 | (14.0) |
| > $80 000 | 475 | (49.0) |
| Not reported or prefer not to answer | 93 | (9.6) |
| **Parity (number of previous pregnancies >20 weeks), n (%)** | | |
| 0 (i.e. nulliparity) | 506 | (52.2) |
| 1 | 307 | (31.6) |
| 2 | 95 | (9.8) |
| 3+ | 57 | (5.8) |
| Not reported/Unknown | 5 | (0.5) |
| **Care provider at recruitment, n (%)** | | |
| Obstetrician | 617 | (63.6) |
| Midwife | 135 | (13.9) |
| Family physician | 218 | (22.5) |
| **Smoking, n (%)** | | |
| None | 777 | (80.1) |
| Before this pregnancy | 124 | (12.8) |
| During this pregnancy | 67 | (6.9) |
| Not reported/Unknown | 2 | (0.2) |
| **Chronic health conditions, n (%)** | | |
| Yes | 205 | (21.1) |
| No | 754 | (77.7) |
| Not reported/Unknown | 11 | (1.1) |

*(Continued)*

**Table 1.** (Continued)

| Characteristic | (total n = 970) | |
|---|---|---|
| **BMI, kg/m$^2$, Mean (SD)** | 26.3 | (6.2) |
| **BMI, kg/m$^2$, Median (IQR)** | 24.6 | (21.9 to 29.3) |
| **Prepregnancy BMI, n (%)** | | |
| Underweight (BMI <18.5 kg/m$^2$) | 29 | (3.0) |
| Normal weight (BMI 18.5–24.9 kg/m$^2$) | 493 | (50.8) |
| Overweight (BMI 25.0–29.9 kg/m$^2$) | 232 | (23.9) |
| Obese (BMI ≥30 kg/m$^2$) | 216 | (22.3) |
| **Gestational weight gain, kg, Mean (SD)** | 13.9 | (6.5) |
| **Gestational weight gain, kg, Median (IQR)** | 13.6 | (10.1 to 17.7) |
| **Gestational weight gain, n (%)** | | |
| Below guidelines | 154 | (15.9) |
| Within guidelines | 279 | (28.8) |
| Above guidelines | 537 | (55.4) |

SD, standard deviation; IQR, interquartile range, n, number; BMI, body mass index

Percentages may not total 100 due to rounding.

A number of the novel factors in pregnancy that we explored were significantly predictive of excess weight gain on univariable analysis (S5 File), including: 1) *cognition*, including compensatory health factors such as plans to "eat healthier later" (odds ratio [OR] 1.45, 95% confidence interval [CI] 1.07 to 1.96) and "exercise later" (OR 1.37, 95% CI 1.02 to 1.83), 2) *personality* [25] (agreeableness, OR 1.14, 95% CI 1.00 to 1.30 and conscientiousness, OR 0.86, 95% CI 0.76 to 0.98), and 3) *behaviour* (emotional eating, OR 1.67 95% CI 1.25 to 2.23). In contrast, a number of the novel factors were not associated with excess gain, including 1) plans to "eat less later", 2) pregnancy-related anxiety, 3) personality difficulties with impulse control, perfectionism or emotion suppression, and 4) behavioural factors like eating in the middle of the night.

Nine categories of risk factors which positively and significantly predicted excess pregnancy weight gain were retained as predictors for excess pregnancy weight gain in the final stepwise logistic regression model: nulliparity (adjusted OR [aOR] 1.50, 95% CI 1.04 to 2.16), being overweight before pregnancy (aOR 2.52, 95% CI 1.55 to 4.11), planned pregnancy weight gain above the guidelines (aOR 2.73, 95% CI 1.66 to 4.47), eating in front of a screen (some meals aOR 2.42, 95% CI 1.62 to 3.61 or most meals aOR 2.20, 95% CI 1.27 to 3.81), disagreement with having control of weight gain (aOR 1.88, 95% CI 1.23 to 2.87), perception that family/friends believed pregnant women should eat two times as much as before pregnancy (disagreement, aOR 2.34, 95% CI 1.24 to 4.42, or agreement aOR 3.32, 95% CI 1.54 to 7.14), having difficulties with emotion control (aOR 2.01, 95% CI 1.02 to 3.97), and identifying as being agreeable (aOR 1.31, 95% CI 1.08 to 1.58). Variables which were protective for excess pregnancy weight gain included being underweight pre-pregnancy (aOR 0.23, 95% CI 0.07 to 0.74) and identifying as being conscientious (aOR 0.79, 95% CI 0.66 to 0.95) (Table 2, Fig 2). The Hosmer and Lemeshow goodness-of-fit test produced a p value of 0.519. Regarding discrimination and calibration, the model yielded an AUC of 0.76 (95% CI 0.72 to 0.80; Fig 3), which is moderate [56, 57] and a calibration slope of 0.96 (95% CI, 0.81 to 1.21) (all p < 0.001). In the validation sample, there was a decreased predictive capability (AUC 0.62; 95% CI, 0.56 to 0.70; p < 0.001) compared with the AUC from the training sample and the reference model (AUC = 0.50) and a less extreme calibration slope (2.09; 95% CI 1.56 to 2.61).

**Table 2. Factors which predict excess pregnancy weight gain in a prospective cohort study.**

| Predictors | aOR (95% CI) | *P* value |
|---|---|---|
| **Parity** | | |
| Nulliparous | 1.50 (1.04 to 2.16) | 0.031 |
| Multiparous | 1.0 (reference) | NA |
| **Prepregnancy BMI** | | |
| Underweight | 0.23 (0.07 to 0.74) | 0.014 |
| Normal weight | 1.0 (reference) | NA |
| Overweight | 2.52 (1.55 to 4.11) | <0.001 |
| Obese | 1.38 (0.83 to 2.30) | 0.217 |
| **Frequency of eating in front of a screen** | | |
| None or almost no meals | 1.0 (reference) | NA |
| Some meals | 2.42 (1.62 to 3.61) | <0.001 |
| Most meals or more | 2.20 (1.27 to 3.81) | 0.005 |
| **Planned pregnancy weight gain** | | |
| Not reported | 1.06 (0.50 to 2.24) | 0.885 |
| Within guidelines | 1.0 (reference) | NA |
| Below guidelines | 0.67 (0.43 to 1.04) | 0.075 |
| Above guidelines | 2.73 (1.66 to 4.47) | <0.001 |
| **Whether my weight changes is up to me** | | |
| Disagree or strongly disagree | 1.88 (1.23 to 2.87) | 0.003 |
| Neither disagree nor agree | 1.42 (0.92 to 2.20) | 0.115 |
| Agree or strongly agree | 1.0 (reference) | NA |
| **Think that family and friends believe that pregnant women need to eat two times as much as before pregnancy** | | |
| Disagree or strongly disagree | 2.34 (1.24 to 4.42) | 0.009 |
| Neither disagree nor agree | 1.0 (reference) | NA |
| Agree or strongly agree | 3.32 (1.54 to 7.14) | 0.002 |
| **When I'm upset, I know I can find a way to eventually feel better** | | |
| Most of the time or almost always | 1.0 (reference) | NA |
| About half the time | 2.01 (1.02 to 3.97) | 0.044 |
| Almost never or sometimes | 1.97 (1.04 to 3.74) | 0.038 |
| **Agreeable personality** | 1.31 (1.08 to 1.58) | 0.005 |
| **Conscientious personality** | 0.79 (0.66 to 0.95) | 0.011 |

aOR, adjusted odds ratio; CI, confidence interval; NA, not applicable; BMI, body mass index.

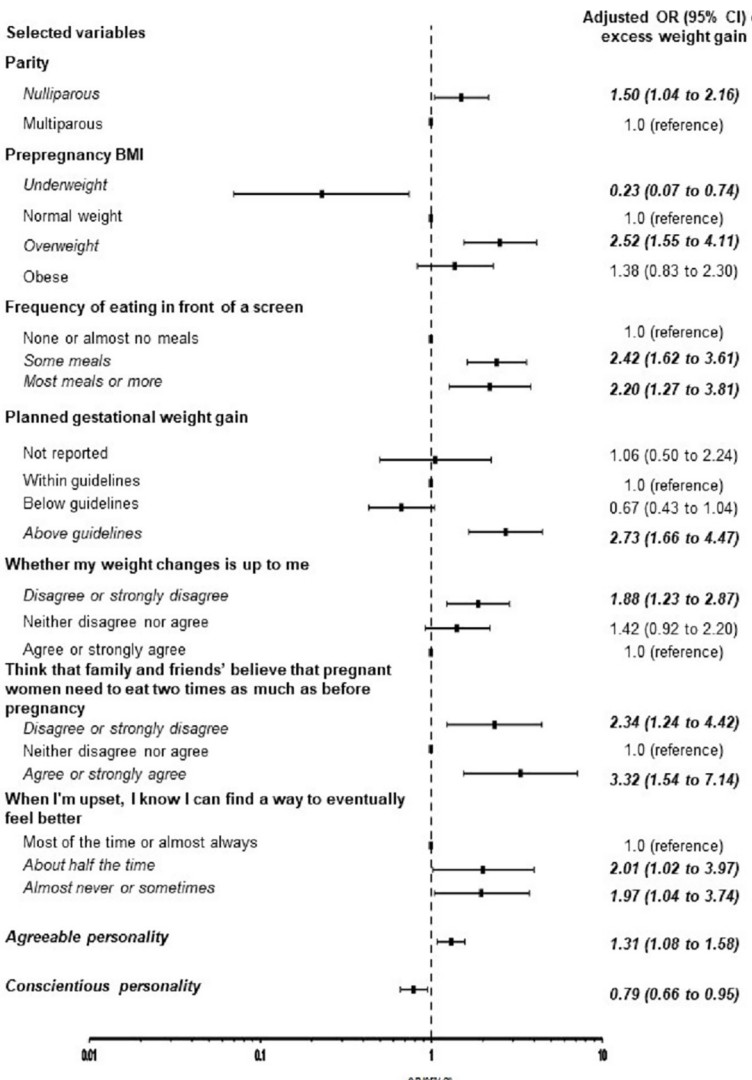

**Fig 2. Final model of adjusted odds of excess pregnancy weight gain in prospective cohort to develop a prediction model.** BMI, body mass index; CI, confidence interval; OR, odds ratio.

In the sensitivity analyses, we found some selected predictors in common, although some differed, and these analyses yielded comparable predictive powers as that derived from the whole sample (S6 File). We focused on the analysis derived from the whole sample because: 1) in early pregnancy, we do not yet have the information to determine if a woman will gain weight above, within, or below the recommendations; and 2) the IOM guidelines recommend that both adults and adolescents should gain weight within the recommendations.

## Discussion

Our validated model that predicted excess pregnancy weight gain included the variables: nulliparity, being overweight, planning excessive gain, eating in front of a screen, low self-efficacy regarding pregnancy weight gain, thinking family or friends believe pregnant women should eat twice as much as before pregnancy, being agreeable, and having emotion control difficulties.

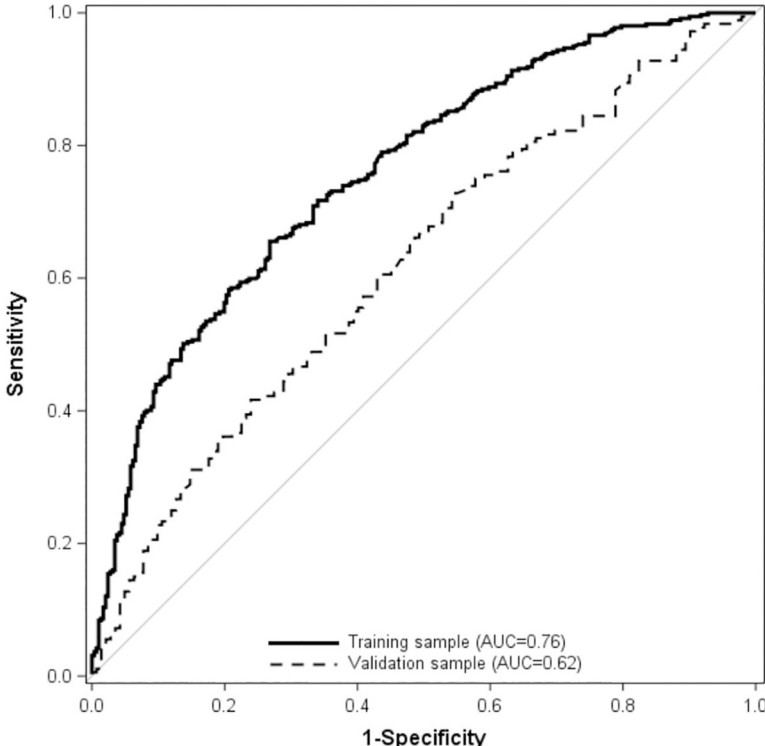

**Fig 3. Receiver operator curves of performance of prediction model of excess pregnancy weight gain.** The AUC in the training set was 0.76, 95% CI 0.72 to 0.80) and in the testing set (0.62, 95% CI, 0.56 to 0.70) p < 0.001. AUC, area under curve.

To our knowledge, this is the first study to explore the prediction of excess pregnancy weight gain with factors ascertained during pregnancy with validation of the model, a key step in any prediction model (literature search in S7 File). [29] Although numerous studies have explored factors associated with excess pregnancy weight gain, despite the fact that some studies have used the word prediction in their titles and manuscripts, the only other validated model relied on *pre*-pregnancy dual-energy X-ray absorptiometry scans in 63 women. [58]

This was the first antenatal study to examine weight gain in relation to personality, a key psychological domain. Individuals with conscientious personalities show a preference for dependable, self-disciplined, and planned behaviours. Agreeable individuals are sympathetic and warm, and value getting along with others. [25] We identified that "agreeable" women were at risk for excess gain while "conscientious" women were protected. Other more easily modifiable predictors of excess pregnancy weight gain were identified. Eating in front of a screen [59] can contribute to distracted eating, and predicted excess gain. Planning to gain weight in excess of the recommendations is associated with excess gain, as we [39] and others [60] have previously shown; although, this is the first prospective study examining this factor. Women assume pregnancy weight gain is not important if is not mentioned by their clinician. [61] Although clinicians may be hesitant to address weight, 84% of pregnant women endorsed being 'comfortable or very comfortable' discussing weight and weight gain with their care provider. [62] Women's thoughts that family/friends' believe that pregnant women should eat twice as much as before pregnancy predicted excess gain. Women who disagreed with this statement were also at increased risk of excess gain, possibly because they believe that women should eat more than twice as much. Although addressing psychological issues may seem

daunting for most pregnancy care providers, this can be operationalized by beginning with educating women on appropriate pregnancy weight gain, the risks of eating "for two", and instead the benefits of eating "twice as healthy". Brief motivational interviewing [63] has shown promise in addressing a variety of health behaviours outside of pregnancy, including sedentary behaviour and body weight, and contraceptive use, [64] and also pregnancy weight gain, [65] for which it has been endorsed by some regional perinatal health services. [66] Women with low self-efficacy over their pregnancy weight gain had substantially higher risks of gaining above the guidelines, however, self-efficacy in pregnancy has been successfully improved in randomized trials. [67]

Strengths of our study include its assessment of both novel psychological factors and ones proven to be associated with pregnancy weight gain in systematic reviews of the literature, [20, 68] and the validation of our predictive model. Our study provides new insights into factors involved in pregnancy weight gain. Our data included validated psychological scales whenever possible. One scale, i.e. family and friends' attitudes toward pregnancy weight gain, was developed by the investigators based on the study of Hales et al. [69] We had a low rate of missing data. We addressed missing data (<3% for most variables) using multiple imputations. We also compared the final group analyzed to groups missing data of major characteristics and study outcomes, and found that they were similar. To increase generalizability, we recruited women from obstetricians, family physicians, and midwives, and from both community clinics and academic centres from across Ontario and from clinics located in more and less socioeconomically disadvantaged areas. Other strengths include the relatively robust size of our cohort, based on an a priori sample size calculation. Our study's limitations include participants with relatively high socioeconomic status, although rates of excess gain were in keeping with the province's population-based data. [2] Although evidence from systematic reviews and a pilot study guided selection of possible predictors by experts in obstetrics, midwifery, family medicine, and psychology, our final model's area under the curve was 0.76, considered "acceptable" or "moderate" discrimination. [56, 57] The AUC derived from the testing sample was lower than that of the training sample, as is common, but at a level suggesting further investigation is needed regarding the better prediction of excess pregnancy weight gain. Future investigation is needed to assess whether the addition of some genetic or biochemical markers may improve the predictive power compared with the questionnaire-based model. Prepregnancy body weight and height were self-reported, which might have resulted in misclassification of prepregnancy BMI for some study participants. However, these are what is used in the clinical context as care providers ask those two questions of pregnant women as a standard part of antenatal care.

In this first validated prediction model in pregnancy, we found that excess pregnancy weight gain was moderately predicted by nine psychological, physical, and social factors. This research highlights the importance of a number of the predictors including relatively easily modifiable ones such as appropriate plans for weight gain and mindfulness during eating. This research moves the field from association studies to prediction and provides an important foundation for future prediction models for excess pregnancy weight gain, an epidemic affecting more than half of women and their infants. Future studies should also be prospective, include predictions with validation, and explore novel factors, such as food as a reward or biomarkers to better predict excess GWG, which adversely impacts the health of mothers and infants.

## Supporting information

**S1 File. Previously explored and novel factors in pregnancy for gestational weight gain.** (DOCX)

**S2 File. Questionnaire to develop a prediction model of excess pregnancy weight gain.**
(PDF)

**S3 File. Comparison of characteristics between women with and without missing data.**
BMI = body mass index; n = number; SD = standard deviation; wk = weeks; yr = years.
(DOCX)

**S4 File. Differences in exposure variables among women by pregnancy weight gain status.**
Data are means (standard deviation) and number of participants (percentage). Percentages
may not total 100 due to rounding.BMI, body mass index; TPB, theory of planned behavior.
(DOCX)

**S5 File. Univariable logistic regression analysis of predictors of excess pregnancy weight gain.** BMI, body mass index; HCP, health care provider; TPB, theory of planned behavior.
(DOCX)

**S6 File.** Sensitivity analyses using samples restricted to A) women with age > 19 years old and B) women without inadequate weight gain.
(DOCX)

**S7 File. Literature search for prediction model for GWG.**
(DOCX)

**S8 File. Protocol for prospective cohort pregnancy weight gain study.**
(DOCX)

## Author Contributions

**Conceptualization:** Sarah D. McDonald, Sherry van Blyderveen, Louis Schmidt, Wendy Sword, Meredith Vanstone, Anne Biringer, Helen McDonald.

**Data curation:** Zhijie Michael Yu.

**Formal analysis:** Zhijie Michael Yu, Joseph Beyene.

**Funding acquisition:** Sarah D. McDonald, Joseph Beyene.

**Methodology:** Sarah D. McDonald, Zhijie Michael Yu, Sherry van Blyderveen, Louis Schmidt, Wendy Sword, Meredith Vanstone, Anne Biringer, Helen McDonald, Joseph Beyene.

**Software:** Zhijie Michael Yu.

**Supervision:** Sarah D. McDonald.

**Writing – original draft:** Sarah D. McDonald.

**Writing – review & editing:** Sarah D. McDonald, Zhijie Michael Yu, Sherry van Blyderveen, Louis Schmidt, Wendy Sword, Meredith Vanstone, Anne Biringer, Helen McDonald, Joseph Beyene.

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
