## [Decision Letter · Decision Letter 0]

17 Mar 2020

PONE-D-19-31116

Prediction of excess pregnancy weight gain using psychological, physical, and social predictors: a validated model in a prospective cohort study

PLOS ONE

Dear Dr. McDonald,

Thank you for submitting your manuscript to PLOS ONE. After careful consideration, we feel that it has merit but does not fully meet PLOS ONE’s publication criteria as it currently stands. Therefore, we invite you to submit a revised version of the manuscript that addresses the points raised during the review process.

We would appreciate receiving your revised manuscript by May 01 2020 11:59PM. To enhance the reproducibility of your results, we recommend that if applicable you deposit your laboratory protocols in protocols.io, where a protocol can be assigned its own identifier (DOI) such that it can be cited independently in the future. For instructions see: http://journals.plos.org/plosone/s/submission-guidelines#loc-laboratory-protocols

We look forward to receiving your revised manuscript.

Kind regards,

Dayana Farias, Ph.D

Academic Editor

PLOS ONE

Journal Requirements:

"The Hamilton Research Ethics Board (REB #13-021) as well as local sites’ Research Ethics Board approval was obtained prior to study initiation.".

i) Please amend your current ethics statement to include the full name of the ethics committee/institutional review board(s) that approved your specific study.

ii) Once you have amended this/these statement(s) in the Methods section of the manuscript, please add the same text to the “Ethics Statement” field of the submission form (via “Edit Submission”).

Reviewers' comments:

Reviewer's Responses to Questions

**Comments to the Author**

1. Is the manuscript technically sound, and do the data support the conclusions?

Reviewer #1: No

Reviewer #2: Yes

2. Has the statistical analysis been performed appropriately and rigorously? 

Reviewer #1: No

Reviewer #2: Yes

3. Have the authors made all data underlying the findings in their manuscript fully available?

Reviewer #1: Yes

Reviewer #2: Yes

4. Is the manuscript presented in an intelligible fashion and written in standard English?

Reviewer #1: Yes

Reviewer #2: Yes

5. Review Comments to the Author

Reviewer #1: Prediction of excess pregnancy weight gain using psychological, physical, and social predictors: a validated model in a prospective cohort study

The study aimed to develop and validate a predictive model of excess pregnancy weight gain using psychological, physical, ad social determinants of excess weight gain collected in early pregnancy.

Although prediction studies are scarce in the gestational weight gain literature, there are issues in this paper that were not properly addressed and it needs to be improved to be considered for publication.

General comments:

Introduction:

In general, it needs to be revised. Some major points:

1. Line 55. Reference 5 do not support the sentence because refers to only one country. A better reference to be explored in this section would have been Scott C, Andersen CT, Valdez N, et al. No global consensus: a cross-sectional survey of maternal weight policies. BMC Pregnancy Childbirth. 2014;14:167. Published 2014 May 15. doi:10.1186/1471-2393-14-167.

2. Line 59. The authors mention “hundreds of studies have examined factors associated with weight gain in pregnancy’ and refer to a meta-analysis related to parity only. There is a need to incorporate more information regarding other associated factors, not only the psychological ones that were the focus of the second paragraph.

Materials and methods:

1. The approach used for the prediction modelling is quite old and more modern techniques (machine learning) should have been applied (a good example in the field would be: Weber A, Darmstadt GL, Gruber S, et al. Application of machine-learning to predict early spontaneous preterm birth among nulliparous non-Hispanic black and white women. Ann Epidemiol. 2018; 28(11): 783 – 789. e1. doi:10.1016/j.annepidem.2018.08.008)

2. Also, the authors decide to work with GWG categorized. It would be interesting to look at it as a continuous variable first, before using the IOM recommendations to categorize it. Also, joining insufficient and adequate weight gain should be avoided. Maybe a sensitive analysis comparing women with excessive vs. adequate weight gain (removing insufficient) could be performed to ensure the joining of the two categories is not introducing bias.

3. The assumption of a 2kg weight gain in the first trimester seems high for some women. Several scenarios (0, 0.5, 1, 1.5, 2 kg) could have been tested, in a sensitivity analysis.

5. Lines 161-171. The calculation of the sample size is out of order and should come right after line 103. The ideas presented in this section are a bit confusing with the present order.

6. The inclusion of the predictors could have happened in groups and there are other approaches to evaluate the fit of the model and the selection of variables. A good example would be: de Freitas Ferreira M, de Moraes CL, Braga JU, Reichenheim ME, da Veiga GV. Abusive alcohol consumption among adolescents: a predictive model for maximizing early detection and responses. Public Health. 2018;159:99–106. doi:10.1016/j.puhe.2018.02.008.

7. The decision for the collection of some variables is not clear at all (e.g.: ‘planning excessive gain’).

8. What was the imputation model considered?

Results:

1. Table 1 is a chart, not a table.

2. Numerical variables (e.g.: BMI, weight gain) could be presented as means, SDs (or medians and interquartile ranges).

Discussion:

In general, the discussion is very superficial, lacks content and debate regarding other factors (the focus is on the psychological ones even though the aim is to evaluate a broader range of factors).

1. What is ‘agreeable’ and ‘conscientious’? The authors use this terminology without a brief explanation to enlighten the reader.

2. It is not correct to say that the comparison between the groups with and without missing data addresses selection bias. Please, review the sentence. This comparison is merely an attempt to support the assumption of MCAR or MAR (in this case, since multiple imputation was used, MAR) and does not address selection bias at all! In the best-case scenario, if the assumption of MAR holds and the amount of missing is not elevated, multiple imputation can help addressing that type of bias.

3. The comparison with reference 64 is irrelevant. The models and outcomes are completely different, and this is not a basis for comparison with these results. If studies with GWG are scarce, these is worth mentioning (and not comparing GWG models with diabetes!!).

4. Finally, what is the applicability of the prediction model? The authors declare in the introduction that the study could ‘provide direction for future interventions’, but they do not discuss in which sense the findings could help to provide those directions.

Minor comments

- Line 85. Was it a problem with the reference?

- Was the sample restricted to adults? This is not mentioned.

- Line 127. The word ‘age’ is missing

- Was the final weight measurement obtained at the day of delivery? If not, there should be a limit because a weight measured 1 month prior to birth (or even before) could not be used for total weight gain calculation (it simply does not reflect the total!).

- How were the means of the imputed datasets calculated? By using Rubin’s rules? If so, it is worth mentioning.

- Line 285. The rate of follow-up mentioning is irrelevant if longitudinal data was not used.

- References need to be formatted according to the rest of the text (remove hyperlink from reference manager)

Reviewer #2: The aim of this study was to develop and validate a prediction model for excess pregnancy weight gain using early pregnancy factors which future interventions might address.

In my opinion the study is interesting and well designed. The sample size is appropriate and the conclusions are drawn based on the data presented. The authors provided all data underlying the findings described in their manuscript as supporting information of the manuscript.

I have only one remark. Authors should include in the study limitations information that height and weight were self-reported, not measured. This could be the reason for incorrect BMI classification before pregnancy of some participants.

6. PLOS authors have the option to publish the peer review history of their article (what does this mean?). If published, this will include your full peer review and any attached files.

Reviewer #1: No

Reviewer #2: No

---

## [Author Response · Author response to Decision Letter 0]

26 Mar 2020

We have submitted the following responses to reviewers as a file and have also copied them below.

Thank you for the reviewers’ comments. We have responded to each below.

Reviewer #1: Prediction of excess pregnancy weight gain using psychological, physical, and social predictors: a validated model in a prospective cohort study

The study aimed to develop and validate a predictive model of excess pregnancy weight gain using psychological, physical, ad social determinants of excess weight gain collected in early pregnancy.

Although prediction studies are scarce in the gestational weight gain literature, there are issues in this paper that were not properly addressed and it needs to be improved to be considered for publication.

General comments:

Introduction:

In general, it needs to be revised. Some major points:

1. Line 55. Reference 5 do not support the sentence because refers to only one country. A better reference to be explored in this section would have been Scott C, Andersen CT, Valdez N, et al. No global consensus: a cross-sectional survey of maternal weight policies. BMC Pregnancy Childbirth. 2014;14:167. Published 2014 May 15. doi:10.1186/1471-2393-14-167.

Response: 

Thank you for the suggestion. The reference has been included.

2. Line 59. The authors mention “hundreds of studies have examined factors associated with weight gain in pregnancy’ and refer to a meta-analysis related to parity only. There is a need to incorporate more information regarding other associated factors, not only the psychological ones that were the focus of the second paragraph.

Response: 

Thank you for the suggestion. We included additional references detailing the risk factors related to excess gestational weight gain, as below:

1. O'Brien EC, Alberdi G, McAuliffe FM. The influence of socioeconomic status on gestational weight gain: a systematic review. Journal of Public Health. 2018 Mar 1;40(1):41-55.

2. Muktabhant B, Lawrie TA, Lumbiganon P, Laopaiboon M. Diet or exercise, or both, for preventing excessive weight gain in pregnancy. Cochrane Database of Systematic Reviews. 2015(6).

Materials and methods:

1. The approach used for the prediction modelling is quite old and more modern techniques (machine learning) should have been applied (a good example in the field would be: Weber A, Darmstadt GL, Gruber S, et al. Application of machine-learning to predict early spontaneous preterm birth among nulliparous non-Hispanic black and white women. Ann Epidemiol. 2018; 28(11): 783 – 789. e1. doi:10.1016/j.annepidem.2018.08.008)

Response: 

We chose the standard logistic regression method as it is the most commonly used approach in clinical research. We agree with the reviewer that machine learning algorithms could be used, and we plan to develop such prediction models in future studies, although we also note that a recent systematic review found that machine learning was not superior to traditional logistic regression in clinical research (Christodoulou E, Ma J, Collins GS, Steyerberg EW, Verbakel JY, Van Calster B. A systematic review shows no performance benefit of machine learning over logistic regression for clinical prediction models. Journal of Clinical Epidemiology. 2019 Jun;110:12–22.). Starting with the traditional analytic method, logistic regression, will allow future comparison of the performance accuracy of the two approaches. 

2. Also, the authors decide to work with GWG categorized. It would be interesting to look at it as a continuous variable first, before using the IOM recommendations to categorize it. Also, joining insufficient and adequate weight gain should be avoided. Maybe a sensitive analysis comparing women with excessive vs. adequate weight gain (removing insufficient) could be performed to ensure the joining of the two categories is not introducing bias.

Response: 

Thank you for your comments and suggestions. The primary aim of this study was to develop a prediction mode for excess gain during early pregnancy. Treating gestational weight gain categorically is a standard approach suggested by both the IOM and other studies, as per these references

1. Institute of Medicine (US) and National Research Council (US) Committee to Reexamine IOM Pregnancy Weight Guidelines. Weight gain during pregnancy: reexamining the guidelines. Rasmussen KM, Yaktine AL, editors. Washington (DC): National Academies Press (US); 2009. (The National Academies Collection: Reports funded by National Institutes of Health). 

2. Dzakpasu S, Fahey J, Kirby RS, Tough SC, Chalmers B, Heaman MI, et al. Contribution of prepregnancy body mass index and gestational weight gain to adverse neonatal outcomes: population attributable fractions for Canada. BMC Pregnancy Childbirth. 2015 Feb 5;15:21. 

In future studies, we will look at predictors of gestational weight gain treated as a continuous outcome. We agree with the reviewer, this is of scientific interest. Nonetheless, due to the complexity of the study design and analytic work, this is beyond the scope of this study.

The primary aim of this study is to develop a prediction model for excess gestational weight gain using early pregnancy factors among women who are pregnant. It may compromise the predictive power and the application of a model if the model was developed using the data restricted to a subsample of the study participants. 

3. The assumption of a 2kg weight gain in the first trimester seems high for some women. Several scenarios (0, 0.5, 1, 1.5, 2 kg) could have been tested, in a sensitivity analysis.

Response: 

We evaluated our data and found that among 364 women who had first trimester weight gain measured between 12-13 weeks of gestation, the average weight gain (mean ± SD) was 1.83 ± 3.3 kg. Thus, our assumption of 2 kg is in accord with this. Second, 2 kg is a standard cutoff used by researchers in this field, as is recommended by the IOM guidelines. 

1. Institute of Medicine (US) and National Research Council (US) Committee to Reexamine IOM Pregnancy Weight Guidelines. Weight gain during pregnancy: reexamining the guidelines. Rasmussen KM, Yaktine AL, editors. Washington (DC): National Academies Press (US); 2009. (The National Academies Collection: Reports funded by National Institutes of Health). 

2. Dzakpasu S, Fahey J, Kirby RS, Tough SC, Chalmers B, Heaman MI, et al. Contribution of prepregnancy body mass index and gestational weight gain to adverse neonatal outcomes: population attributable fractions for Canada. BMC Pregnancy Childbirth. 2015 Feb 5;15:21. 

5. Lines 161-171. The calculation of the sample size is out of order and should come right after line 103. The ideas presented in this section are a bit confusing with the present order.

Response: 

We appreciate this suggestion and moved the sample size calculation, as suggested.

6. The inclusion of the predictors could have happened in groups and there are other approaches to evaluate the fit of the model and the selection of variables. A good example would be: de Freitas Ferreira M, de Moraes CL, Braga JU, Reichenheim ME, da Veiga GV. Abusive alcohol consumption among adolescents: a predictive model for maximizing early detection and responses. Public Health. 2018;159:99–106. doi:10.1016/j.puhe.2018.02.008.

Response: 

Thank you for the suggestion. We chose the standard logistic regression approach as it is the most commonly used approach in clinical research. However, our future studies will implement additional methods, including the bootstrapping method adapted in the recommended study.

7. The decision for the collection of some variables is not clear at all (e.g.: ‘planning excessive gain’).

Response: 

We developed the survey questionnaires based on our systematic review, a pilot study, other studies, and in consultation with five experts with related content expertise. Regarding the Reviewer’s concerns on the health care providers’ recommendations on weight gain, we added the following description as included in the supplementary file:

We also collected data on the participants’ recollection of the health care providers’ recommendations on GWG, including their recommended first trimester weight gain, total GWG, and planned GWG.

8. What was the imputation model considered?

Response: 

Thank you for the opportunity to clarify our description on the imputation methods. We used the fully conditional specification method to address missing data. The method uses the regression method for all imputed continuous variables and the discriminant function method for all imputed classification variables. For each imputed variable, all other variables are used as the covariates. The method gives the most plausible estimates of the missing predictor data, given the data of the available predictors. Therefore, the resulting data are less biased than those derived from the completed-case-analysis. We described the imputation method used to address missing values in the methods section, as below:

We addressed missing data with multiple imputations using the fully conditional specification method to create ten imputed data sets[42] with PROC MI in SAS.[43, 44]

Results:

1. Table 1 is a chart, not a table.

Response: 

We apologize that we do not understand this comment; Table 1 is a table not a chart.

2. Numerical variables (e.g.: BMI, weight gain) could be presented as means, SDs (or medians and interquartile ranges).

Response: 

We appreciate this suggestion and have added means (SDs) and medians (IQRs) for all numerical variables in Table 1.

Discussion:

In general, the discussion is very superficial, lacks content and debate regarding other factors (the focus is on the psychological ones even though the aim is to evaluate a broader range of factors).

1. What is ‘agreeable’ and ‘conscientious’? The authors use this terminology without a brief explanation to enlighten the reader.

Response: 

We added descriptions of the two personality traits in the relevant discussion section (please see below).

Individuals with conscientious personalities show a preference for dependable, self-disciplined, and planned behaviours. Agreeable individuals are sympathetic and warm, and value getting along with others. 

2. It is not correct to say that the comparison between the groups with and without missing data addresses selection bias. Please, review the sentence. This comparison is merely an attempt to support the assumption of MCAR or MAR (in this case, since multiple imputation was used, MAR) and does not address selection bias at all! In the best-case scenario, if the assumption of MAR holds and the amount of missing is not elevated, multiple imputation can help addressing that type of bias.

Response: 

We agree and have revised the statement accordingly (please see below).

We addressed missing data (<3% for most variables) using multiple imputations. We also compared the final group analyzed to groups missing data of major characteristics and study outcomes, and found that they were similar.

3. The comparison with reference 64 is irrelevant. The models and outcomes are completely different, and this is not a basis for comparison with these results. If studies with GWG are scarce, these is worth mentioning (and not comparing GWG models with diabetes!!).

Response: 

We agree and deleted this statement.

4. Finally, what is the applicability of the prediction model? The authors declare in the introduction that the study could ‘provide direction for future interventions’, but they do not discuss in which sense the findings could help to provide those directions.

Response: 

Thank you, we removed ‘provide direction for future interventions’ from the introduction. In the conclusions section, we discussed the future directions with the following sentence: 

Future studies should also be prospective, include predictions with validation, and explore novel factors, such as food as a reward or biomarkers to better predict excess GWG, which adversely impacts the health of mothers and infants.

Minor comments

- Line 85. Was it a problem with the reference?

Response: 

The number listed is the identification number provided by the Hamilton Research Ethics Board - REB #13-021. 

- Was the sample restricted to adults? This is not mentioned.

Response: 

The youngest study participants were 17 years old. We collected data for pregnant individuals and did not restrict data collection to adults.

- Line 127. The word ‘age’ is missing

Response: 

Thank you for pointing this out, we added ‘age’ in the sentence.

- Was the final weight measurement obtained at the day of delivery? If not, there should be a limit because a weight measured 1 month prior to birth (or even before) could not be used for total weight gain calculation (it simply does not reflect the total!).

Response: 

The final weight measurement was obtained before delivery. The average gestational age at the time of the final weight measurement was 38.6 ± 1.6 weeks. The median (IQR) was 38.9 (38.0, 39.7) weeks. We calculated the rate of weight gain per week during the second and third trimesters, as suggested by the IOM recommendations, and compared those rates to the rates in the guidelines.

- How were the means of the imputed datasets calculated? By using Rubin’s rules? If so, it is worth mentioning.

Response: 

We computed the arithmetic means from the ten imputed values. As we had a large sample size in this study, our results derived from this method would thus likely be comparable to those using Rubin’s rules.

- Line 285. The rate of follow-up mentioning is irrelevant if longitudinal data was not used.

Response: 

We agree and deleted this statement.

- References need to be formatted according to the rest of the text (remove hyperlink from reference manager)

Response: 

Thank you, we have formatted the references and removed the hyperlink from the list.

Reviewer #2: The aim of this study was to develop and validate a prediction model for excess pregnancy weight gain using early pregnancy factors which future interventions might address.

In my opinion the study is interesting and well designed. The sample size is appropriate and the conclusions are drawn based on the data presented. The authors provided all data underlying the findings described in their manuscript as supporting information of the manuscript.

I have only one remark. Authors should include in the study limitations information that height and weight were self-reported, not measured. This could be the reason for incorrect BMI classification before pregnancy of some participants.

Response: 

We appreciate the comments and suggestions and added the limitation, as below:

Prepregnancy body weight and height were self-reported, which might have resulted in misclassification of prepregnancy BMI for some study participants. However, these are what is used in the clinical context as care providers ask those two questions of pregnant women as a standard part of antenatal care.

---

## [Decision Letter · Decision Letter 1]

7 Apr 2020

PONE-D-19-31116R1

Prediction of excess pregnancy weight gain using psychological, physical, and social predictors: a validated model in a prospective cohort study

PLOS ONE

Dear Dr. McDonald,

Thank you for submitting your manuscript to PLOS ONE. After careful consideration, we feel that it has merit but does not fully meet PLOS ONE’s publication criteria as it currently stands. Therefore, we invite you to submit a revised version of the manuscript that addresses the points raised during the review process.

We would appreciate receiving your revised manuscript by May 22 2020 11:59PM. To enhance the reproducibility of your results, we recommend that if applicable you deposit your laboratory protocols in protocols.io, where a protocol can be assigned its own identifier (DOI) such that it can be cited independently in the future. For instructions see: http://journals.plos.org/plosone/s/submission-guidelines#loc-laboratory-protocols

We look forward to receiving your revised manuscript.

Kind regards,

Dayana Farias, Ph.D

Academic Editor

PLOS ONE

Reviewers' comments:

Reviewer's Responses to Questions

**Comments to the Author**

1. If the authors have adequately addressed your comments raised in a previous round of review and you feel that this manuscript is now acceptable for publication, you may indicate that here to bypass the “Comments to the Author” section, enter your conflict of interest statement in the “Confidential to Editor” section, and submit your "Accept" recommendation.

Reviewer #1: (No Response)

Reviewer #2: All comments have been addressed

2. Is the manuscript technically sound, and do the data support the conclusions?

Reviewer #1: Yes

Reviewer #2: Yes

3. Has the statistical analysis been performed appropriately and rigorously? 

Reviewer #1: Yes

Reviewer #2: Yes

4. Have the authors made all data underlying the findings in their manuscript fully available?

Reviewer #1: No

Reviewer #2: Yes

5. Is the manuscript presented in an intelligible fashion and written in standard English?

Reviewer #1: Yes

Reviewer #2: Yes

6. Review Comments to the Author

Reviewer #1: I’d like to thank the authors for answering the comments and incorporating the suggestions to the text whenever possible.

I strongly recommend that better prediction models are used in the future, but I agree with the answers provided to the comments on these models. I still have some considerations, though.

a. I strongly recommend that the sample is restricted to adults (Age >19 yo). The use of IOM recommendations and WHO cutoffs for pregnant adolescent women is debatable, so removing adolescents from the sample should be considered.

b. I reinforce the need to compare excessive gestational weight gain using as reference the ‘adequate’ category, not the combination of insufficient + adequate. This would be a sensitive analysis only, to enrich the results and the sample size would not be substantially reduced since only 15% of women presented weight gain below the guidelines.

c. If the 2-kg subtraction for first trimester weight was based on the results from the data, this should be added to the text. The IOM recommendations for first trimester vary between 0.5 – 2kg, so any amount in this interval could have been used. The justification for 2-kg was according to the pattern observed in a smaller sample of women and should be informed in the text, as it was in the answer to my comments.

d. The Introduction section is still focusing on psychological factors, not mentioning and exploring the other factors that might be associated with weight gain and were included in the models.

e. The fact that the AUC for the training set was low (< 0.70) is a problem the authors do not explained well enough. This low AUC may suggest that the prediction model is not good enough and its generalizability may be compromised. What are the solutions for that? What is the possible explanation?

Reviewer #2: All comments have been addressed, thus I am fully satisfied with the authors' response. The authors provided all data underlying the findings described in their manuscript.

7. PLOS authors have the option to publish the peer review history of their article (what does this mean?). If published, this will include your full peer review and any attached files.

Reviewer #1: No

Reviewer #2: No

---

## [Author Response · Author response to Decision Letter 1]

24 Apr 2020

(Our resposne to reviewers has also been submitted as a file.)

Dear Dr. Joerg Heber,

We appreciate the comments from the reviewers and have responded to each one below.

Review Comments to the Author

Reviewer #1: I’d like to thank the authors for answering the comments and incorporating the suggestions to the text whenever possible.

I strongly recommend that better prediction models are used in the future, but I agree with the answers provided to the comments on these models. I still have some considerations, though.

a. I strongly recommend that the sample is restricted to adults (Age >19 yo). The use of IOM recommendations and WHO cutoffs for pregnant adolescent women is debatable, so removing adolescents from the sample should be considered.

Response: 

Thank you for this comment. As suggested, we performed the sensitivity analyses restricting the study sample to women aged 20 years and older. We found that this model has five predictors in common with the one derived from the whole sample. Although these models have some predictors in common and other predictors differ, they have comparable predictive powers. 

In addition, the Institute of Medicine (IOM) weight gain guidelines (1) and the ACOG committee opinion, (2) and the Health Canada (3) suggest that the recommendations can be applied to both adults and adolescents. 

Further, the IOM, on page 3-4, states "Evidence available since the IOM (1990) report is also insufficient to continue to support a modification of the GWG guidelines for adolescents (< 20 years old) during pregnancy. The committee also determined that prepregnancy BMI could be adequately categorized in adolescents by using the WHO cutoff points for adults, in part because of the impracticality of using pediatric growth charts in obstetric practices." 

On page 251, it states “As discussed in Chapter 4, the committee was unable to identify sufficient evidence to continue to support a modification of the GWG guidelines for adolescents (females <20 years old) (Vishwanathan et al., 2008) (see Chapter 4)."

1. Institute of Medicine (US) and National Research Council (US) Committee to Reexamine IOM Pregnancy Weight Guidelines. Weight gain during pregnancy: reexamining the guidelines. Rasmussen KM, Yaktine AL, editors. Washington (DC): National Academies --Press (US); 2009. (The National Academies Collection: Reports funded by National Institutes of Health). 

2. American College of Obstetricians and Gynecologists. ACOG Committee opinion no. 548: weight gain during pregnancy. Obstet Gynecol. 2013 Jan;121(1):210–2. 

3. Canada H. Prenatal Nutrition Guidelines for Health Professionals: Gestational Weight Gain [Internet]. aem. 2010 [cited 2020 Apr 22]. Available from: https://www.canada.ca/en/health-canada/services/can

We, therefore, reported the results of the whole sample. We added a supplementary file reporting these data (S6 File). In statistical analyses section, we added the following statement:

We performed sensitivity analyses by restricting the study sample to women who were aged 20 years and older and separately to women who gained weight within and above the recommendations. 

At the end of the Results, we added the following statement to explain the reasons to use the whole sample but not the restricted samples:

In the sensitivity analyses, we found some selected predictors in common, although some differed, and these analyses yielded comparable predictive powers as that derived from the whole sample (S6 File). We focused on the analysis derived from the whole sample because: 1) in early pregnancy, we do not yet have the information to determine if a woman will gain weight above, within, or below the recommendations; and 2) the IOM guidelines recommend that both adults and adolescents should gain weight within the recommendations.

b. I reinforce the need to compare excessive gestational weight gain using as reference the ‘adequate’ category, not the combination of insufficient + adequate. This would be a sensitive analysis only, to enrich the results and the sample size would not be substantially reduced since only 15% of women presented weight gain below the guidelines.

Response: 

We performed the sensitivity analyses requested and found that excluding women who gained weight below the recommendations yielded a model with the predictive power comparable with the one derived from the whole sample. Although some predictors differed, there were some selected predictors in common among the models derived from the whole sample and the sample without inadequate weight gain. We planned to focus on prediction of excess GWG in early pregnancy, and in early pregnancy there is no way to know whether women will gain above, within, or below the recommendations. We, therefore, reported the results of the whole sample. We added a supplementary file reporting these data (S6 File). In the statistical analyses and results, we added the statements to explain the rationales why we report the results of the whole sample, as shown in our responses to the first comments. 

c. If the 2-kg subtraction for first trimester weight was based on the results from the data, this should be added to the text. The IOM recommendations for first trimester vary between 0.5 – 2kg, so any amount in this interval could have been used. The justification for 2-kg was according to the pattern observed in a smaller sample of women and should be informed in the text, as it was in the answer to my comments.

Response: 

We apologize for not adding the first-trimester weight gain data in the manuscript and added the following statement into the result section.

The mean first trimester weight gain measured between 12-13 weeks of gestation was 1.83 ± 3.3 kg.

d. The Introduction section is still focusing on psychological factors, not mentioning and exploring the other factors that might be associated with weight gain and were included in the models.

Response: 

We apologize for this and have added some statements regarding planned weight gain, parity, and prepregnancy BMI in the introduction, as below:

In addition to psychological factors, our pilot study found that planning gaining weight above the recommendations was associated with the increased likelihoods of developing excess pregnancy weight gain compared with panning weight gain within the recommendations. Other studies also reported that some physical factors, including parity and prepregnancy BMI, might play a role in excess pregnancy weight gain development.

e. The fact that the AUC for the training set was low (< 0.70) is a problem the authors do not explained well enough. This low AUC may suggest that the prediction model is not good enough and its generalizability may be compromised. What are the solutions for that? What is the possible explanation?

Response: 

We agree with the reviewer that the AUCs of our prediction model were moderate, i.e., 0.76 for training and 0.62 for testing datasets, respectively. We added some statements in limitations regarding the improvement of the predictive power in future studies.

Future investigation is needed to assess whether the addition of some genetic or biochemical markers may improve the predictive power compared with the questionnaire-based model.

Reviewer #2: All comments have been addressed, thus I am fully satisfied with the authors' response. The authors provided all data underlying the findings described in their manuscript.

Response: 

We thank the reviewers for reviewing our manuscript and appreciate these comments.

---

## [Decision Letter · Decision Letter 2]

13 May 2020

Prediction of excess pregnancy weight gain using psychological, physical, and social predictors: a validated model in a prospective cohort study

PONE-D-19-31116R2

Dear Dr. McDonald,

We are pleased to inform you that your manuscript has been judged scientifically suitable for publication and will be formally accepted for publication once it complies with all outstanding technical requirements.

With kind regards,

Dayana Farias, Ph.D

Academic Editor

PLOS ONE

Reviewers' comments:

Reviewer's Responses to Questions

**Comments to the Author**

1. If the authors have adequately addressed your comments raised in a previous round of review and you feel that this manuscript is now acceptable for publication, you may indicate that here to bypass the “Comments to the Author” section, enter your conflict of interest statement in the “Confidential to Editor” section, and submit your "Accept" recommendation.

Reviewer #1: All comments have been addressed

2. Is the manuscript technically sound, and do the data support the conclusions?

Reviewer #1: Yes

3. Has the statistical analysis been performed appropriately and rigorously? 

Reviewer #1: Yes

4. Have the authors made all data underlying the findings in their manuscript fully available?

Reviewer #1: Yes

5. Is the manuscript presented in an intelligible fashion and written in standard English?

Reviewer #1: Yes

6. Review Comments to the Author

Reviewer #1: Thank you for kindly answering my comments and addressing the most critical issues. The research was well-conducted and the results are very interesting.

7. PLOS authors have the option to publish the peer review history of their article (what does this mean?). If published, this will include your full peer review and any attached files.

Reviewer #1: No